# A fragment-based approach identifies an allosteric pocket that impacts malate dehydrogenase activity

Atilio Reyes Romero[1], Serjey Lunev[2], Grzegorz M. Popowicz[3,4], Vito Calderone[5 ✉], Matteo Gentili[6], Michael Sattler [3,4], Jacek Plewka[7], Michał Taube [8], Maciej Kozak [8,9], Tad A. Holak [7], Alexander S. S. Dömling [1] & Matthew R. Groves [1 ✉]

Malate dehydrogenases (MDHs) sustain tumor growth and carbon metabolism by pathogens including *Plasmodium falciparum*. However, clinical success of MDH inhibitors is absent, as current small molecule approaches targeting the active site are unselective. The presence of an allosteric binding site at oligomeric interface allows the development of more specific inhibitors. To this end we performed a differential NMR-based screening of 1500 fragments to identify fragments that bind at the oligomeric interface. Subsequent biophysical and biochemical experiments of an identified fragment indicate an allosteric mechanism of 4-(3,4-difluorophenyl) thiazol-2-amine (4DT) inhibition by impacting the formation of the active site loop, located >30 Å from the 4DT binding site. Further characterization of the more tractable homolog 4-phenylthiazol-2-amine (4PA) and 16 other derivatives are also reported. These data pave the way for downstream development of more selective molecules by utilizing the oligomeric interfaces showing higher species sequence divergence than the MDH active site.

[1] Drug Design, University of Groningen, Department of Pharmacy, Groningen, The Netherlands. [2] EV Biotech, Zernikelaan 8, Groningen, the Netherlands. [3] Institute of Structural Biology, Helmholtz Zentrum München, Neuherberg, Germany. [4] Department of Chemistry, Technical University of Munich, Garching, Germany. [5] CERM and Department of Chemistry, University of Florence, Sesto Fiorentino, Italy. [6] Giotto Biotech S.r.l, Sesto F.no, Italy. [7] Faculty of Chemistry, Jagiellonian University, Krakow, Poland. [8] Department of Macromolecular Physics, Faculty of Physics, Adam Mickiewicz University, Poznań, Poland. [9] National Synchrotron Radiation Centre SOLARIS, Jagiellonian University, Kraków, Poland. ✉email: calderone@cerm.unifi.it; m.r.groves@rug.nl

Malate dehydrogenases are a group of multimeric enzymes, typically self-organized as dimers or tetramers, that reversibly catalyze the oxidation of malate to oxaloacetate using the reduction of NAD+ to NADH as a cofactor[1]. Based on subcellular localization, eukaryotic cells possess two isoforms (cytoplasmic and mitochondrial), though yeasts possess a third in glyoxysomes[2]. While cytoplasmic malate dehydrogenase (MDH) supports the malate–aspartate shuttle—allowing reducing equivalents to pass through the inner mitochondrial membrane—mitochondrial MDH is involved in NADH and citrate production to support the electron-transport chain within the TCA cycle[1]. Under physiological conditions, MDH regulates essential biochemical pathways related to energy metabolisms, like gluconeogenesis and the fatty acid cycle, and cell division. In many forms of breast cancer, the biomass growth depends on NAD regeneration by lactate dehydrogenase, an oxidoreductase enzyme structurally related to MDH. Thus, an high LDH expression represents a diagnostic hallmark in cancer[3]. However, in the presence of LDH antagonists like oxamate, cancer cells adopt an evasion mechanism involving MDH which is recruited in order to support glycolysis in the absence of oxygen and production of NAD[4]. This suggests that a tailored therapy aimed at modulating this supportive role of MDH within the glycolysis could hinder the abnormal energetic need required by cancer for proliferation[5]. Several antagonists are reported for one of the two MDH isoforms or both. For instance, elevated expression of HsMDH2 is associated with prostate cancer development and chemotherapy resistance[6] and, to date, 5-benzylpaullones remains the most potent compound for selective inhibition of this specific isoform, as exemplified by inhibitor **4k**[7]. Dual target inhibition becomes particularly effective when both HsMDH1 and HsMDH2 are overexpressed in NSCLC patients[8]. In this regard, trimethylpentane derivatives were identified and characterized as potent HsMDH1/MDH2 dual-target inhibitors[9]. Specifically, compound **16c** demonstrated the highest efficacy in a mouse xenograft assay and inhibited mitochondrial respiration through a competitive inhibition mechanism with NAD.

Given the ubiquity of the dinucleotide binding domain (Rossman fold) in NADH binding enzymes[10] and its presence in 155 reviewed enzymatic activities within human cells (Interpro ID: IPR036291[11]), it is perhaps not surprising that many of these antagonists possess cytotoxic effects. In this regard, non-specific MDH inhibitors are reported in the literature. For instance, while rottlerin, quercetin, and staurosporine aglycone were originally designed for PKCδ/PRAK, PI3-K/GLUT2, and PKC/PKA they also inhibit MDH with an $IC_{50}$ of 0.7, 6.0, and 8.0 μM, respectively[12]. Furthermore, 3 h exposure to 5 μM of the alkaloid staurosporine aglycone led to an 80% decrease in cell viability via caspase activation mechanisms and oxidative DNA damage through reactive oxygen species formation[13,14].

In terms of infectious diseases, both cytoplasmic and mitochondrial MDH have also been found to be anthelmintic targets: Mebendazole hinders the polymerization of microtubules, thus blocking cell divisions. Like the other three known anthelmintics (i.e., Albendazole, Parbendazole, and Thiabendazole), it has been reported that it displays significant inhibitory effects for both MDH and LDH[15,16]. In a similar fashion, MDH was also suggested to be the drug target of miconazole, econazole, and sulconazole, originally designed and approved as antifungal medication for local use with unprecedented inhibitory activity (95–99% motility inhibition at 30 μM for 48 h)[17]. Considering that the mechanism of action of these compounds is based on the inhibition of the fungal enzyme 14α-sterol demethylase, these data suggest that they could be repurposed also as MDH inhibitors[18]. This evidence highlight not only the need to discover selective MDH chemical antagonists—leading to more effective treatments—but also that such antagonists should select between the highly conserved active sites of other NAD-dependent enzymes.

Lunev and Batista assessed the role of oligomeric surfaces in regulating the assembly of PfMDH—the deadliest causative agent of Malaria among the five species that affect humans[19]—showing by in vivo and in vitro modulation of its activity that oligomeric surfaces might be targeted for the treatment of diseases related to MDH function[20,21]. Gossypol, a phenol derivative from the cottonseed plant, is the most potent PfMDH inhibitor with an $IC_{50}$ of $2.03 \pm 0.80\,\mu M$[22], followed by oxamic acid derivatives[22]. Interestingly, oxamic acid derivatives display inhibitory effects also for PfLDH, a confirmed antimalaria drug target[23]. PfMDH and PfLDH complement each other to maintain a constant production of NAD, necessary for the oxidation of glucose and biosynthesis of ATP during glycolysis[24] as the treatment of cultures with gossypol caused the reduction of PfLDH expression with concomitant overexpression of PfMDH. Thereby, the design of dual-target inhibitors of both enzymes has been proposed. However, compounds synthesized as potential chemotherapeutics for Malaria suffer from the same low selectivity as those for cancer. The inhibitors **5–7** are also active against both the mammalian mitochondria and cytosolic malate dehydrogenases since the naphthalene ring mimics the planarity of the adenine of NAD and the negative charge from the conjugated oxamate moiety interacted with two highly conserved arginine residues which secure malate in the catalytic pocket[25]. Even gossypol exhibits side effects that may limit its use as an antimalarial compound as it has been initially characterized as a promising male oral contraceptive but later failed due to hypokalemia and the irreversibility of its contraceptive effect[26]. In contrast, much is known about small molecules inhibiting PfLDH, such as azole-based compounds[27], quinoline derivatives[28], tricyclic guanidine derivatives[29], chloroquine[30], and the aforementioned oxamic acid derivatives[22,31].

In summary, while inhibition of MDH presents a major opportunity in both cancer and infectious diseases the currently available modulating molecules all target the NADH binding site, resulting in reduced off-target effects and greatly limiting their clinical usefulness. To address this, we have systematically searched for small molecule fragments that would bind to the oligomeric interface of an exemplar MDH, rather than the active NADH binding site. In support of this approach, an allosteric cryptic site of MDH has been recently proposed[32]. Our search using saturation transfer difference nuclear magnetic resonance (STD-NMR) identified a fragment (4-(3,4-difluorophenyl) thiazol-2-amine; 4DT) for which binding has been orthogonally validated using microscale thermophoresis and a thermal shift assay. We have confirmed the binding site as distant from the active site using X-ray crystallography and show, as hypothesized by us, that such molecules binding at the oligomeric interface inhibit MDH activity. In addition, we present biochemical competition assays of a library of 16 4DT derivatives, which are also screened in an activity assay. Small-angle X-ray scattering assayed the oligomeric state of the protein in solution in the presence of 4DT derivatives, demonstrating that complete disruption of the oligomeric state is not essential for the measured effects. Finally, kinetic data to support the proposed allosteric inhibitory nature of the scaffold 4-phenylthiazol-2-amine is presented. This identification of an allosteric pocket opens the opportunity to overcome the low selectivity associated with substrate or cofactor analogs and presents a starting point for the future development of specific MDH inhibitors.

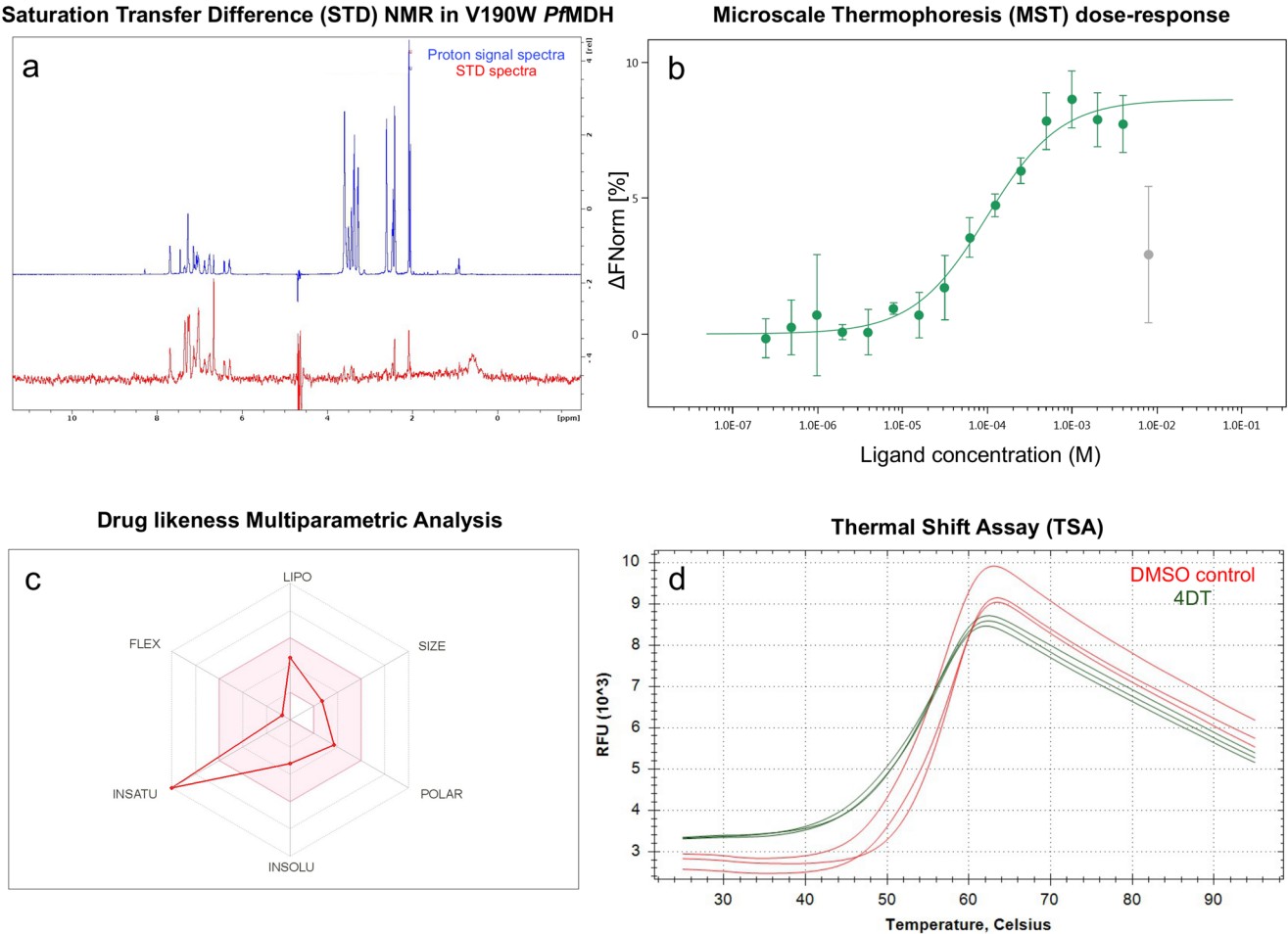

**Fig. 1 Biophysical characterization of 4DT. a** STD-NMR, **b** MST validation with fluorescently labeled WT *Pf*MDH, and **c** drug-likeness profile from SwissADME[61]. The pink area represents the optimal range for each property (lipophilicity: XLOGP3 between −0.7 and +5.0, size: MW between 150 and 500 g/mol, polarity: TPSA between 20 and 130 Å$^2$, solubility: log S not higher than 6, saturation: fraction of carbons in the sp3 hybridization not less than 0.25, and flexibility: no more than 9 rotatable bonds. **d** Thermal destabilization of *Pf*MDH in presence of 4DT (red lines) and control group (green lines) as measured by TSA experiments. RFU relative fluorescence units. Data are represented as mean ± SEM (*n* = 3 biologically independent experiments).

## Results

**Initial identification and orthogonal validation of the fragment hit.** *Pf*MDH WT and V190W were cloned, expressed, and purified with minor modifications[20,33]. V190W *Pf*MDH has been shown to possess a perturbed oligomeric interface resulting in a breakdown of the wild-type tetrameric assembly into soluble dimers with the oligomeric interface being solvent exposed[20]. Totally, 20 mM Tris-base was used in place of 100 mM citrate at the same pH (7.4) and ionic strength (400 mM NaCl). We performed STD-NMR against a library of 1500 fragments from the Maybridge Ro3 diversity fragment library. Initially, 16 fragments were identified to bind to WT *Pf*MDH and 37 to V190W *Pf*MDH. Comparison of the results reveals that seven fragments binding to V190W *Pf*MDH are also bound to WT *Pf*MDH—suggesting 30 potential candidates interacting with the oligomeric interface. Subsequent ranking of these 30 candidates was driven by the principle that the nearer the ligand protons are to the protein, the more likely become highly saturated. Consequently, the signals displaying the strongest intensity in the mono-dimensional STD spectrum reflect the proximity of the ligand to the protein surface[34,35]. Of the 30 fragments, only 4DT showed a high-intensity peak in the aromatic region between 6 and 8 ppm and a weaker intensity peak at 2 ppm (Fig. 1a). Since STD-NMR tells whether or not a fragment binds and in what orientation in the

binding pocket, this was interpreted as indicating that 4DT binds to dimeric V190W *Pf*MDH at a position buried in the WT *Pf*MDH and that the remaining compounds may make a non-specific interaction with the newly exposed tryptophan at position 190. An MST assay orthogonally validated fragment binding with $K_d$ of 99.0 ± 1.7 μM (Fig. 1b) against *Pf*MDH. During the experiment, we observed that saturating concentrations of the compound blenched the fluorescence signal. Nevertheless, no protein aggregates were detected in either of the capillaries. Moreover, 4DT also possesses drug-like properties as shown by the analysis of SwissADMET (Fig. 1c). Finally, 4DT ($T_m = 56 ± 0.17$ °C) destabilizes *Pf*MDH with a measured $\Delta T_m$ of −2.00 ± 0.17 °C with no statistical difference from the control group treated with DMSO ($T_m = 58 ± 0.33$ °C) (Fig. 1d, Supplementary Table 7).

**4-(3,4-difluorophenyl) thiazol-2-amine binds at the oligomeric interface.** In order to elucidate the binding site of 4DT, crystals of *Pf*MDH were soaked in the presence of 2.5 mM of compound, and diffraction data were collected at 2.1 Å resolution (Table 1). The reservoir solution was supplemented with 0.5 mM NADH. The ternary complex with 4DT and NADH is deposited in the PDB under the accession code 6R8G. Based on the presence of

**Table 1 Data collection and refinement statistics.**

|  | 6R8G | 6Y91 |
|---|---|---|
| *Data collection* | | |
| Space group | P2$_1$2$_1$2$_1$ | P2$_1$2$_1$2$_1$ |
| *Cell dimensions* | | |
| *a, b, c* (Å) | 84.76, | 84.62, |
| | 106.89, 145.01 | 107.42, 145.09 |
| *a, b, g* (°) | 90, 90, 90 | 90, 90, 90 |
| Completeness (%) | 99.8 (99.4) | 99.7 (96.6) |
| Multiplicity | 14.2 | 16.8 |
| *I*/σ(*I*) | 14.3 (2.0) | 9.3 (2.1) |
| $R_{merge}$ | 0.19 (0.91) | 0.29 (0.99) |
| *Refinement* | | |
| Resolution (Å) | 2.0 | 2.5 |
| Unique reflections | 89,246 (14220) | 46,208 (3249) |
| $R_{cryst}$/$R_{free}$ | 0.20/0.23 | 0.24/0.28 |
| *No. of atoms* | | |
| Protein | 9566 | 9566 |
| Waters | 320 | 143 |
| Ligand | 130 | 176 |
| *B-factors* | | |
| Protein | 34.2 | 39.1 |
| Waters | 42.4 | 45.4 |
| Ligand | 40.3 | 42.5 |
| RMSD bond lengths (Å) | 0.010 | 0.008 |
| RMSD bond angles (°) | 1.550 | 1.323 |

Values in parenthesis refer to the high-resolution shell (2.23–2.10 Å). †$R_{merge}$ =, where $I_i$(*hkl*) is the mean intensity of the ith observation of symmetry-related reflections *hkl*. ‡$R_{cryst}$ =, where $F_{calc}$ is the calculated protein structure factor from the atomic model ($R_{free}$ was calculated with a randomly selected 5% of the reflections).

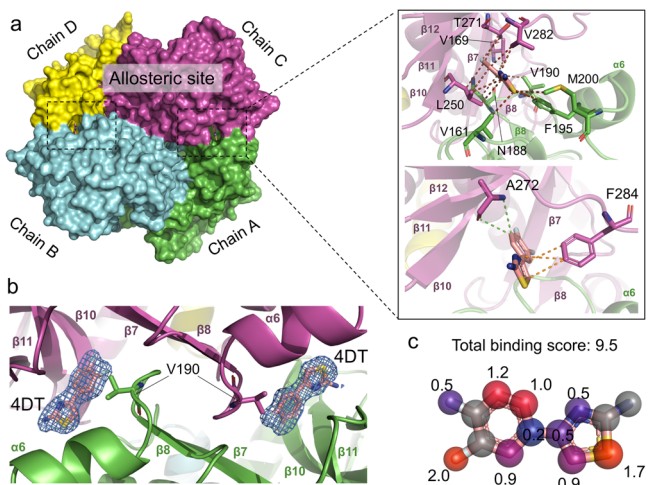

**Fig. 2 4DT bind to an allosteric pocket distinct from the substrate- and NADH-binding sites. a** Left: an overview of WT P*f*MDH tetramer with bound 4DT (pink color, sticks). Monomers are displayed as surfaces. Right: close-up of the binding site of 4DT showing interactions with the protein residues (magenta and green sticks). van der Waals, π-stacking, and other interactions are indicated in brown, orange, and green dotted lines, respectively. **b** Secondary structure representation of the AC interface containing two molecules of 4DT contoured as an isomesh electron density surface (omit map) contoured at 1.0σ. The position of the V190W mutation used for the protein interference method is indicated. **c** Individual atomic scores within the cooperativity binding network. The score is assigned according to an increasing rank, starting with gray (neutral), blue/purple (moderate), and red (relevant).

unambiguous electron density one molecule of NADH and four molecules of 4DT were modeled in the dinucleotide binding site of chain D and at the AC, BD interfaces, respectively. At the AC interface, 4DT occupies a total surface area of 307 Å$^2$, 60% of which consists of contacts with chain A and the remaining 40% with chain C. The fluoride atoms face the β7 sheet, while the amino group and thiazole ring flank the β10–12 sheets from chain A. On the opposite side, these functional groups contact the loops K160–D165 and N188–P192 from chain C (Fig. 2b). PISA analysis quantified the Δ$^i$G P value as equal to −1 for all four molecules, suggesting the interface surface to be interaction-specific rather than an artifact due to crystal contacts. Furthermore, it shows that the binding site mainly consists of hydrophobic contacts. Interestingly, the fragment binds next to V190 whose mutation to tryptophan interferes with the protein native assembly[20]. As shown in Fig. 2a, the benzene and the thiazolyl moieties at the AC interface form a T-shaped π stacking interaction with F284. Further visual inspection of the pocket revealed a wide network of van der Waals interactions between the 4DT sulfur atom and V161, M200, F195, F284 side chains and between the meta fluorine atom and the side chains of V187, N188, and V169. Moreover, the substitution pattern around the benzene ring shows that only the fluorine atom in the meta position possesses a higher score (2.0) than the one in para (0.5). This could be related to the fact that the C chain interface provides two apolar contacts with V190, N188 plus one with V169 of the A chain while at the para position we observed only one between T271 and fluorine. Therefore, a replacement of fluorine with an atom or functional group with a greater Van der Walls radius specifically in meta position could benefit from more steric contacts with the oligomeric interface (Fig. 2c).

**Structural rearrangements related to 4-(3,4-difluorophenyl) thiazol-2-amine binding**. The binding of 4DT did not impact the

overall tetrameric structure of WT P*f*MDH when compared with the apo form (0.47 Å RMSD). Nevertheless, a closer inspection of both the AD and BC interfaces revealed substantial rearrangements of L250 (3.52 Å, CD1), V282 (3.10 Å, CG2), and K273 (2.84 Å, CE) side-chain positions to accommodate 4DT in the binding site (Fig. 3b–e). Moreover, we observed further Cα shifts in the 177–174 loop near the NAD binding site (Fig. 3a). In addition, the atomic coordinates of the loop R75-I87 located between helices α3 and β4 sheet of the catalytic site of chains A–C were omitted due to a lack of electron density, caused by an increase in local structural flexibility. Similarly, disordered active site loops were observed in the NADH bound form and no electron density is observed for residues Q80–L90. Finally, a structural comparison between the ligand and citrate bound forms (Fig. 3f) reveals an active site loop in a solvent-exposed open conformation. In contrast, when P*f*MDH interacts with citrate the active site loop secures the molecule through ionic interactions with arginines R150 and R81 which both confer substrate specificity. This observation was noted also in *Escherichia coli* MDH[36]. Following this loop motion of approximately 80°, the R81 guanidine group loses an ionic contact with the carboxylic functional group of the citrate. Taken together, these data corroborate the existence of an allosteric site in P*f*MDH located at the AD, BC interfaces for the potential development of modulators that do not affect human isoforms as described by Botros et al.[32].

**Sequence and interface analysis of the binding site**. To explore the sequence identities at the 4DT and orthosteric binding sites, we have performed multiple sequence alignment with ClustalW[37] of three MDHs from pathogenic microorganisms and the two human isoforms. P*f*LDH was included in the analysis. This resulted in 97% sequence coverage (Supplementary Figs. 1 and 6) with *Homo sapiens* accounting for the

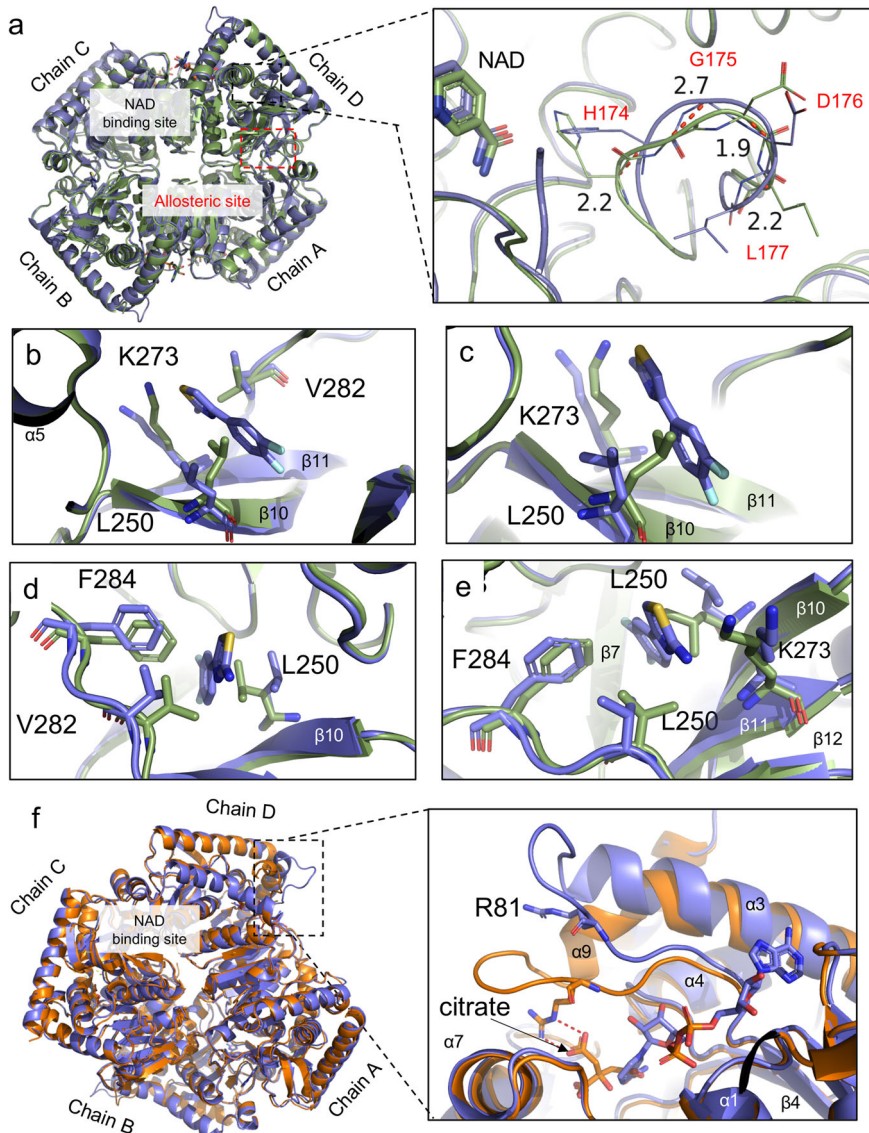

**Fig. 3 Apo (green cartoons), citrate-liganded (orange cartoons, PDB ID:5NFR)[20] and 4DT bound (purple cartoons) structures of *Pf*MDH. a** A close-up view of the active site loop in close proximity to the NAD binding site. Pairwise Cα distances are expressed in Å and represented as red dotted lines while loop amino acids as lines. **b–e** Side chain rearrangements of the interface residues involved in the formation of 4DT binding site (sticks representations). **f** Close-up view displaying the lifting of the active site loop and distancing of R81 (stick representation) with consequent loss of one ionic interaction depicted as red dotted lines.

highest sequence identity (34%) followed by *Bacillus antracis* (28%), *Haemophilus influenzae* (23%), *Staphylococcus aureus* (22%), and lastly *Leishmania major* (21%). Eighteen identical residues (G8, G10, L30, D32, G40, D71, G84, D89, L90, N94, P120, L146, S227, and A233 with D147, R150, R87, and N119) cluster at the N-terminal cofactor binding domain (Rossmann-fold). Five of these (H174, R150, R87, R81, and D147) are involved in the catalytic mechanism where the three arginines interact with the carboxylates groups of the substrate (malate or oxaloacetate) through ionic contacts[38]. Table 2 summarizes the sequence and surface analysis. Overall, the three independent oligomeric interfaces of *Pf*MDH possess a buried solvent accessible surface area of 3481 Å². Furthermore, as shown in Supplementary Fig. 2, five of these interface interactions are specific as they involve residues not strictly conserved in *Pf*LDH or in the MDH *of H. sapiens, B. anthracis, Trypanosoma cruzi* and *Brucella abortus* (V169, N188, V190, F195, and M200) and, only two residues share similar physiochemical

properties with other MDHs (V161 and L250). The AD interface has a relatively small area (361 Å²) and accounts only for 2.5% of the total buried surface area. In contrast, the AB and AC interfaces represent the key contact surface for the oligomerization (1811 and 1309 Å², respectively) and are essential for the biological activity in vitro[20]. In comparison with the other six homologs, the interface residues are less conserved than those of the active site (20%). Like the AD interface, the AC contains no identical residues, however, it accounts for higher buried ASA than the former, making it an attractive target for structure-based drug design. PISA analysis shows that upon ligand binding the guanidine side chain of R183 from chain C loses one salt bridge with the E194 of chain A. Similarly, contacts between K198 and E164 both in common with the citrate and NADH bound form are lost as well as between K228 from chain C and E164 of chain A. These data suggest that 4DT most likely interferes with correct orientation of intra-oligomeric contacts.

**Table 2 Surface analysis with PISA online server[60] and sequence conservation across another malate dehydrogenase.**

| Number of residues | 313 | | | |
|---|---|---|---|---|
| Identical | 18 (5.7%) | | | |
| Similar | 52 (16.6%) | | | |
| Active site residues | 5 | | | |
| Identical | 4 (80%) | | | |
| Similar | 1 (20%) | | | |
| Interface | AB | AC | AD | Total |
| Residues | 50 | 38 | 12 | 100 |
| Identical | 8 (16.0%) | 0 | 0 | 8 (8.0%) |
| Similar | 3 (6.0%) | 7 (18.4%) | 1 (8.3%) | 11 (11.0%) |
| Not identical | 39 (78.0%) | 31 (81.6%) | 11 (91.7%) | 81 (81.0%) |
| Total ASA per monomer (Å$^2$) | 14,577 | | | |
| Buried ASA (Å$^2$) | 1811 (12.4%) | 1309 (9.0%) | 361 (2.5%) | 3481 (23.9%) |

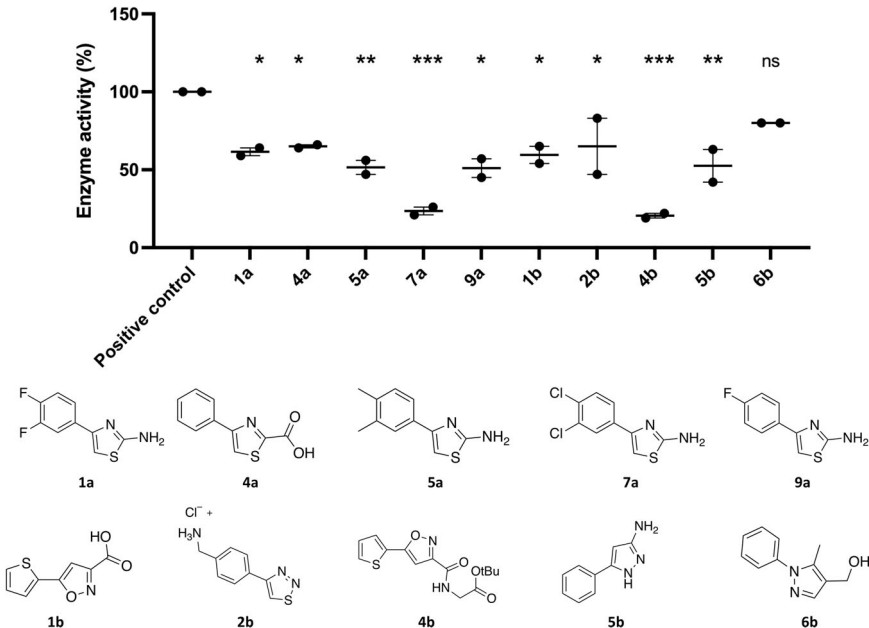

**Fig. 4 Dot plot inhibition of the 4DT derivatives vs. control (DMSO).** Data are represented as mean ± SEM; $*p < 0.05$, $**p < 0.01$, $***p < 0.001$ ($n = 2$ biologically independent experiments).

**4DT derivative screening**. In order to explore the structure–activity relationships of the parental scaffold of 4DT, we purchased new compounds from vendors with a higher electron-withdrawing effect than fluorine (i.e., nitrile and tri-fluoromethyl groups), atomic radius (i.e., chlorine and bromine), and steric effect (i.e., isopropyl). In some cases, the amine group was either replaced with a carboxylic acid or the whole thiazole ring with isoxazole, pyrazole, and thiadiazole. The results of our experiments are presented in Fig. 4 while the ANOVA summary is in Supplementary Table 3. The enzyme activity in the presence of compounds **7a** (23%) and **4b** (20%) was significantly reduced in comparison with the control, followed by compounds **5a** (52%) and **5b** (53%). In addition, enzyme activity measured in the presence of compounds **1a** (62%), **1b** (60%), **2b** (65%), **4a** (65%), and **9a** (51%) resulted in a reduction of *Pf*MDH activity with similar statistical significance (Supplementary Table 4). Only compound **6b** showed no statistical difference with the control. Finally, precipitation of compound was observed for the para-isopropyl (compound **10a**), para-nitro (compound **2a**), meta-trifluoro (compound **8a**), and para-bromo (compound **6a**) deri-vatives, most likely due to their low solubility in the buffer.

The low aqueous solubility of this series of compounds makes the measurement of an accurate IC$_{50}$ curve extremely challenging and the presented reduction in activity should not be used to draw conclusions on the relative inhibitory properties of the compounds.

**Thermal destabilization of 4DT derivatives**. Thermal shift assay (TSA) experiments showed an overall decreased thermal stability of *Pf*MDH. The molecule with a trifluoromethyl group in meta position ($T_m = 48.0 \pm 1.0$ °C) (Supplementary Fig. 3a) displayed a statistically significant decrease compared with control ($\Delta T_m = -10$ °C), suggesting that bulky functional groups can thermically destabilize in a more efficient manner the protein as a result of the steric effect at the *Pf*MDH A/C interface. In contrast, as can be seen in Supplementary Fig. 3b, c, the presence of the com-pounds 4PA ($T_m = 55.0 \pm 0.2$ °C) as well as its substituted dimethyl derivative ($T_m = 54.0 \pm 0.2$ °C) shows a decrease in melting temperature of $\Delta T_m = -3.0$ °C and $-4.0$ °C, respectively with no statistical difference from the control ($T_m = 58.0 \pm 0.3$ °C) (Supplementary Table 7).

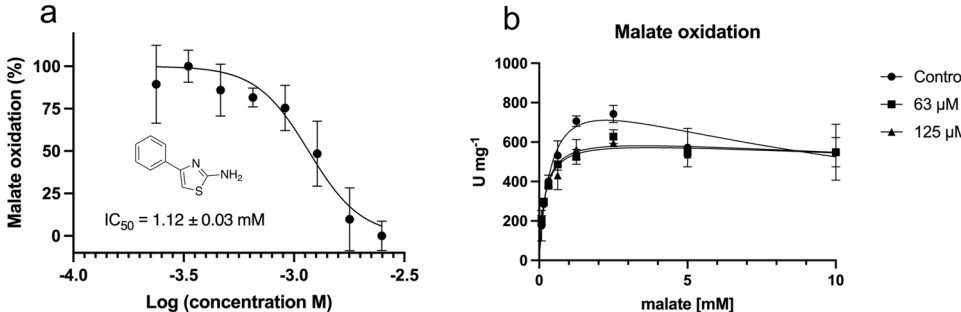

**Fig. 5 Study of the biological response of _Pf_MDH in the presence of 4PA. a** Left part: dose/response inhibition of P*f*MDH. Data are presented as mean ± SEM (_n_ = 3 biologically independent experiments). **b** Kinetics studies of two different concentrations of 4PA vs. control (DMSO) using the substrate inhibition model. Experiments were carried in the same plate. _y_-Axis shows enzyme activity expressed as U = μmol of NAD+ reduced per minute by 1.0 mg of _Pf_MDH. Data are presented as mean ± SEM (_n_ = 3 biologically independent experiments).

**Inhibitory and allosteric nature of 4PA**. Experiments using 4PA and performed in 5 mM L-malate revealed an IC$_{50}$ value of 1.17 ± 0.04 mM (Fig. 5a, Supplementary Table 5). We observed turbidity in the treated wells, most likely due to the low compound solubility in the chosen buffer. Nevertheless, lower doses show the disappearance of the effect, thus confirming the inhibitory nature of 4PA. Based on this observation, we decreased the 4PA concentration. Subsequent kinetic measurements were performed with _Pf_MDH, varying the concentration of malate as the reaction substrate in optimal assay conditions. The results are presented in Fig. 5b and the model fitting summary in Supplementary Table 6. In comparison to other reports, the control showed substrate inhibition at the highest malate concentration (10 mM)[24]. Thus, the oxidation of the malate curve was fitted using a substrate inhibition model, resulting in a calculated $V_{max}$ of 967.8 ± 99.0 U mg$^{-1}$ and $K_m$ of 0.4 ± 0.1 mM. The same experiment was performed in the presence of 63 μM 4PA which showed a decreased $V_{max}$ of 645.7 ± 29.0 U mg$^{-1}$ and $K_m$ of 0.2 ± 0.0 mM. Similarly, doubling the concentration of 4PA to 125 μM led to $V_{max}$ of 630.2 ± 28.5 U mg$^{-1}$ and $K_m$ of 0.2 ± 0.003 mM. Moreover, our data indicate a decrease in substrate binding affinity induced by the presence of 4PA with a calculated $K_i$ of 12.5 ± 4.2 mM in the control and $K_i$ 63.0 ± 31.8 mM and 72.9 ± 41.9 mM in the presence of 63 and 125 μM 4DT, respectively. Taken together, these data indicate a non-competitive mechanism of the inhibition of 4PA and a simultaneous decrease in the binding affinity for malate.

**Small-angle X-ray scattering**. In order to further validate the interference effect at the oligomeric interface, we performed small-angle X-ray scattering (SAXS) experiments. SAXS is routinely used to determine the oligomeric states of proteins in solution due to the noninvasive character of this technique. As SAXS detects even nanoscale electron density fluctuations it can be employed to determine the tetrameric and dimeric states of malate dehydrogenase. Only the 4DT derivatives that previously showed no aggregation in DLS, i.e., **2a** and **6a**, were tested with SAXS since aggregated protein samples give SAXS signals that cannot be interpreted. As can be seen in Supplementary Fig. 4a, b, the samples of _Pf_MDH apoprotein and in complex with compound **2a** appear to be tetramers of ca. 110 kDa (expected molecular weight calculated from the sequence would be 136 kDa), and the radius of gyration of 33.8 Å (Supplementary Table 8). Only the complex with compound **6a** seems slightly more open (Supplementary Fig. 4c), with an Rg of 36.4 Å. Finally, Fig. 6 shows that the three scattering curves fit the crystal structures of _Pf_MDH in the absence and presence of compounds **2a** and **6a** well with the goodness-of-fit below 1.0 as calculated

using crysol software[39], thus suggesting a good correlation between crystal structures and complexes in a solution.

## Discussion

MDH is involved in energy metabolism, pyrimidine biosynthesis as well as supplying the TCA cycle, sustaining abnormal cell growth like in cancer or parasite infection like in _P. falciparum_. Hence, the _Hs_MDH1 and _Hs_MDH2 antagonists were synthesized (Supplementary Table 1) for cancer, or substrate/cofactor analogs, i.e., oxamic acid derivatives for _Pf_LDH (Supplementary Table 2) and _Pf_MDH for parasitic diseases. Unfortunately, they lack on-target selectivity, since they bind a strongly conserved active site. Allosteric modulation of this class of enzymes may allow an improved selectivity between host and pathogen MDHs, as well as between MDHs and LDHs, thus leading to better-tolerated treatments.

_Pf_MDH represents a model protein well studied by us both in vitro[20] and in vivo[21]. The surface analysis results show that 16% of AB interface residues are identical across six exemplificative species (i.e., _H. influ_enzae, _B. antracis_, _H.sapiens_, _L. major_, _S. aureus_, _B. antracis_) and _E. coli_ dimeric MDH (29% sequence identity)[20]. In contrast, the AC interface has a slightly higher number of non-identical residues than AB (81.6% vs. 78%), but no identical ones. These data indicate that, firstly, the AC interface could be targeted by small molecules with potentially improved specificity and, secondly, each MDH possesses a unique AC/BD network of contacts. Moreover, the hypothesis of the existence of an allosteric site has been advanced by mutagenic studies, yet remained cryptic until now[32]. We have screened for mutations on this interface and could only identify the V190W as a mutation that resulted in a soluble dimer that retained the active site architecture. For example, the V190I mutation resulted in an insoluble construct, and mutations at other points of the dimeric interface did not have the desired effect of impacting the oligomeric state of _Pf_MDH. Therefore, we engineered a soluble mutant (V190W) that does not exist in nature, with a solvent-exposed oligomeric surface for comparative analysis of hits from NMR screening.

Our joint experiments based on STD-NMR, X-ray, MST, and TSA led us to elucidate the binding pocket of 4DT, a molecule with potential drug-like properties but still with a weak Kd that nevertheless is typical of fragments[40]. We show that 4DT binds with a Kd of 99 μM to wild-type _Pf_MDH in an MST assay performed using 50 nM _Pf_MDH and a concentration range of 4DT from 240 nM to 4 mM. However, 4DT could not be seen to bind to the same protein under the STD-NMR conditions at a 100-fold molar excess. However, the concentration of protein used in the STD-NMR experiments was 10 μM. This clearly shows the essential problem in high-throughput fragment screening, in that

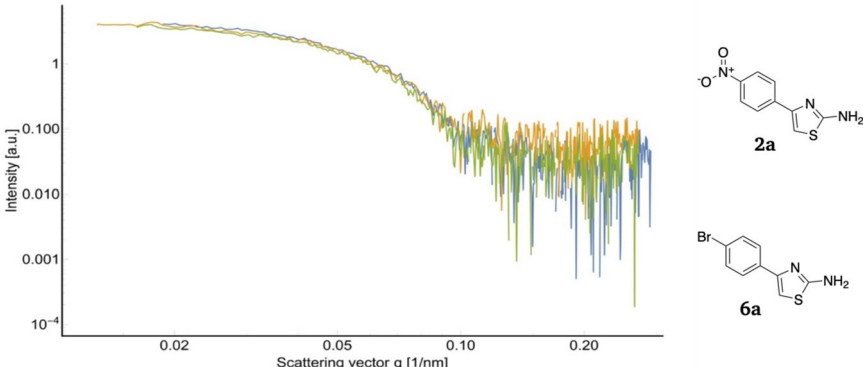

**Fig. 6 Study of oligomeric fluctuations of *Pf*MDH by characterization of its SAXS profile.** The overlay of scattering profiles from *Pf*MDH apoprotein (blue), *Pf*MDH in complex with compound **6a** (orange) and compound **2a** (green).

differing methods often show widely differing Kds. However, we believe that our STD-NMR approach allowed us to identify differential binders, and thereby identify 4DT as a molecule that binds to the oligomeric interface exposed by the V190W mutation.

The allosteric site is located approximately 35 Å away from the malate/cofactor binding site. In this pocket formed 4DT interacts through van der Waals and π-stacking interactions with the surrounding amino acids, including the key residue V190. Upon binding to 4DT, other key residues, for example, L250 and F284, undergo clear structural rearrangements to allocate 4DT with the benzene ring facing the oligomeric interface. Such orientation is in accordance with the NMR spectrum as the high peak intensities can be seen in the aromatic protons. Distal from the allosteric site, we furthermore observed that the active site Q80–L90 loop remains in the open state, ready to allocate a substrate and cofactor molecule. During the refinement, no citrate atoms were modeled due to a lack of consistent electron density, despite the presence of 1.4 M citrate in the reservoir solution. This evidence relates to its effect of protein stabilization as shown by our co-crystal (PDB: 5NFR) and subsequentially increase in $\Delta Tm$ using TSA[20], most likely due to the formation of three ionic bonds, two of them from the active site loop, e.g., R81 and R87, and R150. Thereby, 4DT might reduce the binding affinity towards malate but more generally towards other similar carboxylic acids such as citrate.

At this point, it is tempting to speculate that the binding may well result from the ligand-dependent conformational selection. Due to the small size of the fragments, we cannot exclude the possibility for non-specific binding at other locations of *Pf*MDH. As these fragments are highly hydrophobic in nature it is not unreasonable to suggest also that they may negatively impact the protein fold, without interfering at the dimer interface. To that respect, it should be noticed that all our TSA experiments display a negative $\Delta T_m$ with indeed no statistical difference from the control as for 4DT, 4PA, and compound **5a**. However, only the compound carrying the bulkiest functional group displayed a significant difference suggesting interference with intraoligomeric contacts resembling the indole ring but still not yet strong enough to produce the same effect as the mutation.

To further explore this hypothesis, we have screened 4DT analogs and conducted more in-depth studies with 4PA, a more tractable homolog of 4DT. 4PA and 4DT share the benzene and thiazole rings as this system concentrates the highest cooperative binding score. In addition, these molecules differ only by two fluorines whose van der Waals radius (1.47 Å) is similar to that of a hydrogen atom (1.20 Å). This makes their dimensions very

similar but at the cost of a decreased water solubility for 4DT. Moreover, this compound has been screened for the drug discovery of highly profitable antiviral and anticancer targets like HIV-1 nucleocapsid[41] and the TIR domain of human MyD88[42]. Thus, our choice is motivated as 4PA has proved to be slightly more soluble than 4DT and, due to its structural similarity with 4DT, it served as a model molecule to confirm the allosteric nature of the pocket we elucidated. 4PA displayed both dose–response effect and a $K_i$ decrease by 6–5× fold in presence of 125 and 63 μM of compounds. Taken together, these data indicate the presence of a cryptic allosteric site in *Pf*MDH where 4DT analog may bind in an inhibitory noncompetitive fashion.

These experiments on 4DT certainly cannot exclude a possible interaction with *Hs*MDH1 and *Hs*MDH2 for which it would be necessary to conduct a selectivity study. However, in both cases, the cooperativity binding scores are lower than *Pf*MDH (9.5), meaning that binding to the AC/BD interfaces benefits from the local network of interactions. Besides, we found only 16 crystal structures corresponding to the desired homodimer quaternary structure and none of these is co-crystallized with inhibitors similar to 4DT. This implies that our docking models need to be interpreted cautiously since they cannot be validated with a co-crystallized reference. In that respect, further kinetics experiments should be carried out in the future to better understand the 4DT mechanism of action with both human isoforms.

With the identification of 4DT, a number of 4DT derivatives were screened, and interestingly these did not show the same enzyme activity. For example, in compound **1b** the single removal of a fluorine atom per se does not impact the oxidative activity of *Pf*MDH in the assay. This effect could be traced to the Van der Waals radius of fluorine (1.40 Å), which is similar to that of hydrogen (1.20 Å). Consequently, its substitution with one hydrogen atom does not generate a steric surface at the AC interface. In contrast, replacement with two chlorides of larger Van der Waals radius (1.75 Å) than fluorine significantly impacts enzymatic activity, as shown by derivative **7a**. Nevertheless, we are aware that the diversity of the compound library is insufficient for establishing structure–function relationships models. However, we believe that this evidence suggests bulky functional groups should be preferred for providing more interfering surfaces. With its higher Van der Waals radius (1.85 Å), the para-bromo derivative (**6b**) is an exemplificative case that strengthens the latter hypothesis, as shown by SAXS. Such technique is a great validation technique and as such, it was initially used to check for artificial contacts in the crystal structure of *Pf*MDH as measurements in SAXS are performed in native conditions (no crystal formation, buffer of a choice). As shown in data from software

Crysol, the goodness of fit between SAXS measurement of apo *Pf*MDH and putative one calculated from atom positions in the crystal overlap nicely. Next, we suspected inhibitors to disrupt the quaternary structure of *Pf*MDH into dimers, which SAXS would in theory very nicely follow. It turned out, though, that many of the samples were heavily aggregated preventing SAXS measurements, except for compounds **2a** and **6a**. Here, we found out that compound **6a** opens up the structure of *Pf*MDH as reflected by a higher value for the radius of gyration 36.4 Å compared to 33.7 Å for apo *Pf*MDH meaning that it did not dissociate the complex of 4 *Pf*MDH subunits, but located itself in the middle causing them to distance themselves from each other a bit, which is fascinating given the small size of the compound. The strength of this study lies in the fact that compound **6a** was not forced into the structure of *Pf*MDH due to crystal lattice formation or soaking of the crystal, but consequently bound all (or almost all) available *Pf*MDH under native buffer conditions. Furthermore, we showed that SAXS was able to detect compounds otherwise discarded on the sole basis of their biological activity as exemplified by compound **6b**.

In summary, our data pave the way for downstream development of more selective molecules by utilizing the oligomeric interfaces showing higher species sequence divergence than the MDH active site by using fragment-based screening techniques and activity assay.

## Materials and methods

**Recombinant protein expression and purification.** To obtain the V190W MDH mutant, site-directed mutagenesis was performed as described by Lunev et al.[20]. The full-length WT *Pf*MDH gene was cloned into pETM-13, embedding additional non-cleavable amino acids AAALEHHHHHH at the C-terminus. The construct was verified by automated sequencing (Sanger) (Supporting Information). Subsequently, *E. coli* Rosetta 2 (DE3) pLysS (Novagen) were transformed with a pETM-13 vector bearing the *Pf*MDH gene for recombinant protein production. Cells were grown in TB media supplemented with 50 mg ml$^{-1}$ kanamycin and 35 mg ml$^{-1}$ chloramphenicol. An overnight pre-culture grown at 310 K was used to inoculate the main production culture (supplemented by 0.5% glucose to prevent leaky expression), which was grown at 310 K to an OD 600 of 0.6–0.8. Subsequently, enzyme expression was induced with 1 mM isopropyl-β-D-thiogalactopyranoside, and the cells were incubated for 16–18 h at 310 K. The cells were harvested by centrifugation at 5000 rpm for 30 min at 4 C and resuspended in lysis buffer (50 mM Tris-HCl, 300 mM NaCl, 30 mM imidazole, pH 8.0) protease inhibitors (Complete Mini EDTA-free, Roche Applied Science) and a spatula tip of lysozyme. After sonication on ice, the lysate was centrifuged for 45 min at 18,000 rpm and the supernatant was passed through a 0.45 μm filter (Whatman) to remove traces of unlysed cells and aggregates. The clarified cell lysate was purified by immobilized metal ion affinity chromatography with buffer A (50 mM Tris-HCl, 300 mM NaCl, pH 8.0) and buffer B (50 mM Tris-HCl pH 8.0, 300 mM NaCl, 500 mM imidazole) over a 5 mL His-Trap HP column (GE Healthcare). Bound protein was eluted from the column at 125–150 mM imidazole. The sample obtained after Ni-NTA purification was diluted 6-fold with 50 mM Tris-HCl pH 8.0 and loaded on a 5 ml Cation Exchange column (GE Healthcare). The protein was eluted from the column by a linear gradient ranging from 50 mM to 1 M NaCl. Finally, the sample was concentrated and injected onto a HiLoad 16/600 Superdex 200 column (GE Healthcare), previously equilibrated in buffer C (100 mM Na-phosphate buffer pH 7.4 and 400 mM NaCl). The final yield of pure protein from 1 L culture was typically between 6 and 10 mg ml$^{-1}$ The purity of the protein was assessed by sodium dodecyl sulfate gel as better than 98%.

**NMR experiments.** NMR spectra were acquired on Bruker Avance III (600 MHz) spectrometer at 300 K, equipped with a triple resonance cryoprobe head and an automated SampleJet sample changer. The samples for STD experiments were prepared in 100 mM Na-phosphate pH 7.4, 400 mM NaCl and spectra were recorded in water with 1:100-fold excess of the fragment (dissolved in DMSO at 100 mM) in an additional 10% (v/v) D$_2$O to provide a lock signal. Final protein and ligand concentrations were 10 μM and 1 mM, respectively. Selective saturation of the protein resonances (on resonance spectrum) was performed by irradiating at 0.5 ppm using a series of Eburp2.1000-shaped 90 pulses (50 ms, 1 ms delay between pulses) for a total saturation time of 2.0 s. For the reference spectrum (off-resonance), the samples were irradiated at −10 ppm. On- and off-resonance scans were subtracted using phase cycling. The fragment library screening was performed by testing 300 mixtures containing five fragments each initially. Mixtures that yielded positive STD signals were selected and deconvolved, i.e., fragments that formed

"active" mixtures were tested separately. Spectra were analyzed with TopSpin® software, 4.0

**Microscale thermophoresis.** A Nanotemper MonolithNT.115 instrument (Nanotemper Technologies, GmbH) was used to quantify ligand/protein interactions. Purified *Pf*MDH was labeled using the Red-NHS Monolith Protein Labeling Kit according to the manufacturer's protocol[43]. All reactions were performed in 100 mM Na-phosphate buffer pH 7.4, 400 mM NaCl, 0.05% Tween-20 in standard capillaries (Nanotemper Technologies GmbH). The ligand was 2× fold diluted starting from 4 mM as the highest concentration; the final protein concentration was 50 nM. All samples were incubated with compounds for about 1 h at room temperature followed by centrifugation at 18,000 rpm (Eppendorf Centrifuge 5415R) before loading into capillaries. Measurement was performed in triplicate at 40% LED intensity and 40% MST power. The $K_d$ model was employed to determine the binding constant using the software MO.Affinity Analysis v2.3 while data collection was performed with MO.Control 2 (Nano-temper Technologies, GmbH).

**Thermal shift assay.** TSA is a rapid and economical biophysical technique for protein folding, buffer optimization and it has been revealed a flexible method for high thought screening in early stage drug discovery[44] and fragment-based drug design[45]. The TSA experiment was conducted in a BioRad CSX 96 (Biorad). Each reaction consisted of 1 mM of compound (100 mM stock in 100% (v/v) DMSO) and 10 μM *Pf*MDH in 100 mM Na–phosphate buffer pH 7.4, 400 mM NaCl, and 10× Sypro Orange (Invitrogen) for a total volume of 100 μl. Control experiments were performed with 1% (v/v) DMSO. Inflection points were determined using BioRad CFX96™ control software. Generally, a temperature increase of over 1 °C is interpreted as a result of the stabilization of the protein-ligand interaction. On the contrary, a negative shift is considered a destabilization caused by the fragment.

**Crystallization, X-ray data collection, and processing.** Initial efforts focused on the crystallization conditions described in the literature by Wrenger et al.[33]. Initial hits grew in 22% (w/v) PEG 3350, 4 °C, at a protein concentration of 10 mg ml$^{-1}$ after 2 days. The buffer was supplemented with 500 μM NADH since it increases the thermal stability of the protein[20]. Streak seeding experiments were used to reproduce a large number of crystals in different buffer conditions, though not of sufficient quality for data collection. The crystals were subsequentially optimized by screening the ratio of protein to reservoir solution (1.4 M trisodium citrate, 100 mM Hepes pH 7.5) at 20 °C. First hits appeared in 2–3 days and reached maximum dimension after one week at 10 mg ml$^{-1}$ concentration and a 1:3 ratio of protein: reservoir solution. To generate the complex structure the crystals were then soaked with 2.5 mM ligand for 2 h at 20 °C, transferred into a cryoprotective solution made of reservoir solution supplemented with 30% w/v ethylene glycol, and flash-cooled in nitrogen steam. A dataset was collected in-house, using a D8 Venture diffractometer (BRUKER) equipped with a PHOTON II detector, at 100 K. The crystal diffracted to 2.1 Å resolution and belongs to space group P2$_1$2$_1$2$_1$ with four molecules in the asymmetric unit, corresponding to one functional tetramer, the solvent content of ~43%, and a mosaicity of 0.3°. The data were processed using the program XDS[46], reduced and scaled using XSCALE[46], and amplitudes were calculated using CTRUNCATE[47]. The structure was solved using molecular replacement (MOLREP[48]) with 5NFR as a search model. Refinement was carried out using REFMACS[49] applying TLS restraints. Between refinement cycles, the model was subjected to manual rebuilding using COOT. Water molecules have been added using the standard procedures within the ARP/WARP suite[50]. The quality of the refined structure was assessed using the program MOLPROBITY[51]. Coordinates and structure factors have been deposited at the PDB under the accession code 6R8G (*Pf*MDH in complex with 4DT and NADH) and 6Y91 (*Pf*MDH in complex with NADH).

**Activity assay and data analysis.** The oxidation of malate to oxaloacetate was assayed spectrophotometrically in flat-bottomed Corning 96-well microplates in a total volume of 250.0 μl in 50 mM glycine buffer pH 10.2[24] with an additional 400 mM NaCl for protein stability. The increase of absorbance at 340 nm was recorded in kinetic mode using a Tecan Spark multimode microplate reader at 298 K. A mastermix of 50 mM NAD+ and 44 μM *Pf*MDH was used for both the screening and kinetics experiments. The final concentration of protein and cofactor were 50 and 3 mM, respectively. Positive control groups were set (absence of compound) in the same plate as the treated group. The absorbance from the background noise was subtracted from the data and then a linear regression model was employed in order to get the line slope from the first 5 min of the reaction. The percentage of enzymatic activity has been calculated as described by Eq. (1)

$$\text{enzyme activity} (\%) = \frac{\text{slope treated}}{\text{slope positive control}} * 100 \tag{1}$$

**Enzymatic assay and kinetics experiment.** While the fragment initially discovered was 4-(3,4-difluorophenyl) thiazol-2-amine, we decided to perform

enzymatic and kinetic assays using a similar substructure 4-phenylthiazol-2-amine. 2× fold dilutions from a 200 mM stock solution of the compound in 100% DMSO was prepared for dose–response experiments. 3.3 μL of compound (DMSO in the case of positive control) were mixed with 233.9 μL mastermix of WT $Pf$MDH previously incubated with 3 mM NAD + using a multichannel pipette. The final conditions tested were a maximum of 2.5 mM of 4PA at 1.32% (v/v) final DMSO concentration. The microplate was incubated for 15 min at 25.35 °C and the reaction started with the addition of 13.0 μL of 5 mM L-malate.

Kinetics experiments were performed by mixing 1.3 μL of compound stock solutions from 12.5 and 25 mM of 4PA in 100% DMSO with 223.8 μL mastermix of $Pf$MDH previously incubated in ice with 3 mM NAD+ for 30 min. Final tested concentrations of 4PA were 125.0 and 62.5 μM and 0.5% (v/v) DMSO. Subsequently, eight concentrations of L-malate were prepared apart by 2× fold serial dilution from a 100 mM stock in assay buffer at pH 10.2. After 1 h of incubation at 26 °C, the reaction was started by the addition of 25.0 μL L-malate (5 mM final concentration) with a multichannel pipette. The μmol of NADH produced per minute was extrapolated from the calibration curve and then divided by the corresponding milligrams of the enzyme (0.00175 mg). Kinetic constants $K_m$ (mM) and $V_{max}$ (U * mg$^{-1}$) and $K_i$ (mM) were quantified with GraphPad Prism version 8.0 according to the substrate inhibition Eq. (2)[52].

$$Y = V_{max}^* X / (K_m + X^* (1 + X/K_i)) \quad (2)$$

All measurements were performed in triplicate.

**Screening of 4-phenylthiazol-2-amine derivatives**. 1.3 μL of derivatives of 4-phenylthiazol-2-amine in 100% DMSO were mixed with 237.7 μL of a mastermix of WT $Pf$MDH previously incubated with NAD+. The final compound concentration used in this screening was 500 μM in the presence of 0.5% (v/v) DMSO. The microplate was sealed with silver foil for 1-h incubation at 25–35 °C. Finally, reactions in both the samples and control wells were started upon the addition of 13.0 μL of 5 mM malate). At the end of the experiments, the reaction of the plate was spectrophotometrically assayed at 600 nm in order to identify turbidity due to the low solubility of the compound. All measurements were performed in duplicate.

**Statistics and reproducibility**. Unless otherwise stated, data were presented as the mean ± SEM in triplicate. The normality tests were done with the Anderson-Darling, D'Agostino, Shapiro–Wilk, and Kolmogorov–Smirnov methods. Pairwise $p$ values of 4DT derivative screening were calculated using ordinary one-way ANOVA with Dunnet correction with *$p < 0.05$, **$p < 0.01$, ***$p < 0.001$ indicating the statistical significance. TSA statistics were derived with a non-parametric Kruskal Wallis test. Data were prepared with Microsoft Excel and Graph Pad Prism software (v 8.0).

**Computational analysis**. Molecular interactions as well as the representation of the bond length and the scoring of the cooperativity binding network was generated with Scorpion[53]. Pictures were rendered with PyMol (The PyMOL Molecular Graphics System, Version 2.0 Schrödinger, LLC). The compound 4DT was minimized using OpenBabel[54] software and docked with AutoDock Vina integrated into PyRx[55] using as a receptor either the human mitochondrial $Hs$MDH2 (PDB: 4WLV) or the cytoplasmic homology using pig heart MDH1 as a template (PDB: 4MDH) (Supplementary Methods 1). Homology modeling was performed with SWISS-MODEL (Supplementary Methods 2)[56]. The search volume was composed of the whole protein.

**SAXS experiments**. X-ray scattering experiments were performed using the laboratory SAXS system Xeuss 2.0 (XENOCS, Grenoble, France) equipped with a MetalJet D2 microfocus X-ray generator (0.134 nm wavelength). The protein solution at a concentration of 2.49 mg ml$^{-1}$ was mixed with 4PA derivatives at a concentration of 1 mM, incubated for 1 h at room temperature, and spun at 16,000 rpm (Eppendorf Centrifuge 5415R) to remove any potential aggregates prior to injection into the low-noise liquid sample cell. SAXS data were collected in 3–4 subsequent 10 min long frames using a Pilatus3R 1 M hybrid photon counting detector (Dectris, Switzerland). Data reduction and buffer subtraction procedures were performed using the Foxtrot package. Scattering profiles were assessed for radiation-damaged and averaged. Structural parameters were calculated in Primus from the ATSAS package[57]. The distance distribution function was calculated using GNOM[58].

**Dynamic light scattering**. Measurements were performed on Litesizer 500 (Anton Paar). Samples were incubated with the highest concentrations of compounds for 2 h at room temperature. Prior to measurements, samples were spun for 5 min at 16,000 rpm (Eppendorf Centrifuge 5415R) to remove aggregates.

**Reporting summary**. Further information on research design is available in the Nature Research Reporting Summary linked to this article.

## Data availability

The SAXS data have been deposited in the SASBDB[59] under the access codes SASDLQ2 ($Pf$MDH L-lactate dehydrogenase, apo), SASDLR2 ($Pf$MDH L-lactate dehydrogenase bound to inhibitor **2a**), and SASDLS2 ($Pf$MDH L-lactate dehydrogenase bound to inhibitor **6a**). The structures are currently on hold and will be released upon publication. Coordinates and structure factors have been deposited at the PDB under the accession code 6R8G ($Pf$MDH in complex with 4DT and NADH) and 6Y91 ($Pf$MDH in complex with NADH). The authors confirm that the data supporting the findings of this study are available within the article [and/or] its supplementary materials. Supplementary Data 1: MST data processing (.moc), validation report (.pdf) and raw data used for Fig. 1 (.xlsx). Supplementary Data 2: raw data used for Fig. 1 (.xlsx). Supplementary Data 3.xlsx: PISA alignment and interfaces analysis. Supplementary Data 4.csv: thermal shift assay raw data. Supplementary Data 5.pdf: validation report for Crystal structure of malate dehydrogenase from *Plasmodium falciparum* incomplex with 4-(3,4-difluorophenyl) thiazol-2-amine (PDB ID: 6R8G). Supplementary Data 6.pdf: validation report for the Crystal structure of malate dehydrogenase from *Plasmodium falciparum* in complex with NADH (PDB ID: 6Y91). Supplementary Data 7.xlsx: Statistics and data processing for 4DT derivatives. Supplementary Data 8.xlsx: Statistics and data processing for 4PA dose–response experiment. Supplementary Data 9.xlsx: Statistics and data processing for 4PA kinetics experiments. Supplementary Data 10.doc: C-terminal His-tagged $Pf$MDH WT sequencing data and primary sequence.

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

## Acknowledgements

This project is funded from the European Union's Framework Program for Research and Innovation Horizon 2020 (2014–2020) under the Marie Skłodowska-Curie Grant Agreement No. 675555, Accelerated Early stage drug discovery (AEGIS). M.K. acknowledges also supported by a grant (2017/27/B/ST4/00485) from National Science Centre (Poland). We would also like to thank Dr. Arie Geerlof for his kind support in cloning experiments of *Pf*MDH.

## Author contributions

A.R.R.: performed the experiments (STD, TSA, activity assay, MST, and crystallization), expressed, and purified the protein. Writing—original draft. S.L.: Writing—review and editing. G.M.P.: performed the STD experiment. Supervision. Data analysis. V.C.: performed the experiments (model solving and coordinate deposition in the Protein Data Bank). Data analysis. M.G.: Writing—review and editing, Supervision. M.S.: Writing—review and editing. Supervision. Project administration. J.P.: performed the experiments (SAXS, docking), Writing—review and editing. M.T.: performed the experiments (SAXS). M.K.: performed the experiments (SAXS). T.H.: Writing—review and editing, Supervision. A.D.: Writing—review and editing, Supervision, Funding acquisition. M.R.G.: Conceptualization, Methodology, Writing—review and editing, Supervision, Project administration, Funding acquisition. Investigation. All authors have given approval to the final version of the paper.

## Competing interests

The authors declare no competing interests.

**Additional information**

