## [Peer Review File · Communications Biology]

Reviewers' comments:

Reviewer #1 (Remarks to the Author):

This paper describes the identification of an allosteric pocket on Malate Dehydrogenase (MDH) which has been identified using a fragment-based approach. Malate dehydrogenase has been shown to be an important target in both oncology and in metabolism by pathogens such as *Plasmodium falciparum* (Pf). In this study a fragment library of 1500 fragments were screened against PfMDH. This led to the identification of one fragment, 4DT, which was shown to bind at the oligomeric interface. While the focus of the paper is on this fragment it would have been good to see where the other fragment hits were bound. With the identification of 4DT a number of 4DT derivatives were screened and these did not show the same enzyme activity, in some cases the removal of a single fluorine atom seemed to have a significant negative effect (compound 9a), is there an explanation as to why this is based on the structural data?

While the results in the paper are interesting the research reported is at an early stage and it would have been good to see further optimisation of this fragment. In its current form this paper is not suitable for publications in Nature Communications Biology.

Reviewer #2 (Remarks to the Author):

This is a very interesting manuscript by Romero et al, which provides observation of identifying allosteric inhibitors affecting MH activity. overall, this is a very important study, which will benefit of developing specific inhibitors of MH. There are some points need to be revised before publication.

Major issues

1. This manuscript was not carefully prepared, authors need to make sure the units clearly presented throughout the manuscript, is it μM or mM ?
2. Figure arrangement is very messy, figure 1 appears two times. Please correct them.
3. 4-phenylthiazol-2-amine derivative screening is very useful, but the result was not presented clearly, the presentation of Figure 1 (should be figure 4?) is not convincing. It will be good to have IC_{50} obtained for these derivatives, showing percentage in inhibition can not tell the difference among these compounds.

Minor

1. Line 321, is it V190W or V290W
2. There is no description of the wild type and mutant in Material and methods
3. Are the compounds binding to MH also interacting with other MHs? Although authors have done some docking studies, any biochemical assay or biophysical assay to show the difference in binding to different MHs?

Reviewer #3 (Remarks to the Author):

In this manuscript, the Authors present a fragment based screening strategy whose aim is to identify compounds that will interact at the oligomeric interface of Malate deshydrogenase. They relied on a dimeric mutant and on differential NMR STD to identify such potential fragments. Followingly they characterized on of their best hit by biochemical and biophysical assays, validating an allosteric effect. Finally, using a more soluble analog, they were able to obtain a resolved RX structure of this molecule, displaying an allosteric site. They also further characterize the effect of this 4PA compound and some analogs regarding the activity of pfMDH.

Overall, this study provide one of the rare successful application of the emerging therapeutic trend that is the targeting of oligomeric interfaces. The authors used a complete set of assays, from biophysical and biochemical, to characterize their hits and even obtained a RX structure that will benefit the whole community in the attempt to develop oligomer disruptors. This study is of a major

interest as it paves the way to the extension of druggable target and the development of highly selective compounds. I still have some issues about the data presented in the current manuscript, that I believe need addressing.

General comment :

Something goes wrong with the numerotation of the figures

1. Line 316-317. A general comment, could you state the relevance to target this interface of pfMDH (AC) compared to the other one (AB)?
2. In the section 6.1, line 319 – 322, you stated that you kept molecule that were interacting with the dimeric pfMDH and not with the WT pfMDH (ie tetrameric). But the hit you identified displayed a good affinity (99 μ M) for the WT pfMDH, how do you explain that?
3. For the MST data, initial fluorescence as well as raw thermogram should be provided as supplementary.
4. In your initial statement, line 329, you said that the 4DT was orthogonally validated by MST, but you assayed it only against the WT pfMDH. Did you perform a binding assay on the dimeric pfMDH to verify if its affinity was better than for the tetrameric protein?
5. The DSF data need to have their statistical reproducibility stated, for example SEM for the T_m . Plus, significance should be clarified, ie is a 2°C difference significant? Optionally, is this destabilization dose-dependent?
6. Finally, the start of screening strategy rely on the V190W mutant that is dimeric and you are expecting to identify fragments that bind the very same interface that was mutated. Would not the mutation by a Trp be problematic, ie induce false binding to aromatic compounds? A double control, perhaps with another dimeric mutant on another position would clarify this point and strengthen the NMR screening.
7. Section 6.6. Does some of the compounds induce no effect on the thermal stability? Because it would be strange for even compounds like 2b or 6b that are negative for enzymatic activity to display thermal destabilization.
8. Section 6.7, line 471-474. The statement of the inhibitory nature of 4PA would need to show the dose-dependency on the enzymatic activity. If higher doses are not possible, lower doses would at least show the disappearance of the effect.
9. Section 6.8. I could not understand the statement of this section, what did it bring to the overall study? Plus, compound 2a and 6a are not validated as good analogs of 4DT or 4PA, at least in the data presently displayed.

Dr Maxime Liberelle

Reviewers' comments:

Reviewer #1 (Remarks to the Author):

This paper describes the identification of an allosteric pocket on Malate Dehydrogenase (MDH) which has been identified using a fragment-based approach. Malate dehydrogenase has been shown to be an important target in both oncology and in metabolism by pathogens such as Plasmodium falciparum (Pf). In this study a fragment library of 1500 fragments were screened against PfMDH. This led to the identification of one fragment, 4DT, which was shown to bind at the oligomeric interface. While the focus of the paper is on this fragment it would have been good to see where the other fragment hits were bound.

We appreciate the reviewer's insightful analysis, which has also been raised by referee3. Our approach was a comparative screen of the native tetrameric and an artificially generated dimeric form of PfMDH. While, based on a comparison of the hits identified against both types of PfMDH, 30 compounds could be proposed as selective between the two forms, only a single hit from the compounds screened showed both properties of being selective between the two forms of PfMDH and had experimental data to support a strong binding mode. We have modified the text to clarify this point. (Section 4.1, line numbers 311-319).

With the identification of 4DT a number of 4DT derivatives were screened and these did not show the same enzyme activity, in some cases the removal of a single fluorine atom seemed to have a significant negative effect (compound 9a), is there an explanation as to why this is based on the structural data?

Again, we thank the reviewer for their insight. We believe that at the relatively low affinity shown by 4DT even minor perturbations of the molecules can have significant impacts on their binding affinities. Indeed, an analysis of the atomic cooperativity suggests the importance of this fluorine upon binding. We have added a clarification to the text to address this issue (Section 4.2, line numbers 355 -361).

Reviewer #2 (Remarks to the Author):

This is a very interesting manuscript by Romero et al, which provides observation of identifying allosteric inhibitors affecting MH activity. overall, this is a very important study, which will benefit of developing specific inhibitors of MH. There are some points need to be revised before publication.

Major issues

1. This manuscript was not carefully prepared, authors need to make sure the units clearly presented throughout the manuscript, is it μM or mM ?

We thank the reviewer for the time spent reading and reviewing the manuscript. His/her

observation is correct. We apologize that the submitted version was not carefully prepared. All units of measurement have been corrected.

2. Figure arrangement is very messy, figure 1 appears two times. Please correct them.

We have modified the figure arrangement and hope that it is now clearer.

3. 4-phenylthiazol-2-amine derivative screening is very useful, but the result was not presented clearly, the presentation of Figure 1 (should be figure 4?) is not convincing. It will be good to have IC_{50} obtained for these derivatives, showing percentage in inhibition cannot tell the difference among these compounds.

We appreciate the reviewer's insightful suggestion. These compounds suffer from poor solubility and the generation of a titration curve including high concentrations of the compounds screen suffer from artifacts resulting from this poor solubility. As a result, we believe that the IC_{50} s we have measured would be rather inaccurate. We have modified the text to clarify this point (Section 4.7, 446 – 452).

Minor

1. Line 321, is it V190W or V290W

This should have read "V190W". This typo has been corrected.

2. There is no description of the wild type and mutant in Material and methods

We have updated the missing section with a new paragraph, section 3.1 of Material and Methods (line numbers 132-134). Further details of the sequencing have been added in the SI as .docx file.

3. Are the compounds binding to MH also interacting with other MHs? Although authors have done some docking studies, any biochemical assay or biophysical assay to show the difference in binding to different MHs?

We performed the comparative docking analysis to assess the potential for exploitation of the allosteric pocket we discovered in other MDH exemplars. We are planning to

both elaborate 4DT as a more potent binder of PfMDH, as well as perform the suggest screening against other MDHs. To this point we have no data to support binding of 4DT or derivatives to other MHs.

Reviewer #3 (Remarks to the Author):

In this manuscript, the Authors present a fragment based screening strategy whose aim is to identify compounds that will interact at the oligomeric interface of Malate dehydrogenase. They relied on a dimeric mutant and on differential NMR STD to identify such potential fragments. Followingly they characterized one of their best hit by biochemical and biophysical assays, validating an allosteric effect. Finally, using a more soluble analog, they were able to obtain a resolved RX structure of this molecule, displaying an allosteric site. They also further characterize the effect of this 4PA compound and some analogs regarding the activity of pfMDH.

Overall, this study provide one of the rare successful application of the emerging therapeutic trend that is the targeting of oligomeric interfaces. The authors used a complete set of assays, from biophysical and biochemical, to characterize their hits and even obtained a RX structure that will benefit the whole community in the attempt to develop oligomer disruptors. This study is of a major interest as it paves the way to the extension of druggable target and the development of highly selective compounds. I still have some issues about the data presented in the current manuscript, that I believe need addressing.

General comment :

Something goes wrong with the numerotation of the figures

We thank the reviewer for the time spent reading and reviewing the manuscript. As also indicated by another reviewer, we have corrected the figure numbering and apologise for this error.

1. Line 316-317. A general comment, could you state the relevance to target this interface of pfMDH (AC) compared to the other one (AB)?

We appreciate the reviewer's interest in this topic. In short, a sequence analysis of the residues comprising the AB and AC interface strongly suggested that a significantly larger sequence divergence was available at the AC interface, meaning an improved potential for selectivity between species. We have modified the text to clarify this point in line numbers 524 – 530 (Discussion section).

2. In the section 6.1, line 319 – 322, you stated that you kept molecule that were interacting with the dimeric pfMDH and not with the WT pfMDH (ie tetrameric). But the hit you identified displayed a good affinity (99 μ M) for the WT pfMDH, how do you explain that?

The referee is correct in his/her observation that 4DT can be shown to bind with a Kd of 99 μ M to wild-type PfMDH in an MST assay performed using 50nM PfMDH using a concentration range of 4DT from 240 nM to 4 mM, but could not be seen to bind to the same protein under the STD-NMR conditions at a 100-fold molar excess. However, the

concentration of protein used in the STD-NMR experiments was 10 μ M. This clearly shows the essential problem in high-throughput fragment screening, in that differing methods often show widely differing Kds. However, we believe that our STD-NMR approach allowed us to identify differential binders, and thereby identify 4DT as a molecule that bind to the oligomeric interface exposed by the V190W mutation. The text has been modified to clarify this point (line numbers 539 - 549).

3. For the MST data, initial fluorescence as well as raw thermogram should be provided as supplementary.

We have provided the MST data as requested in the file “MST_report.pdf”

4. In your initial statement, line 329, you said that the 4DT was orthogonally validated by MST, but you assayed it only against the WT pfMDH. Did you perform a binding assay on the dimeric pfMDH to verify if it's affinity was better than for the tetrameric protein?

We appreciate the reviewer's insight. However, the mutation we engineered triggers the opening of the assembly due to the steric hindrance of the indole ring, thus exposing the whole of the oligomeric interface to the solvent. As a result, we have exploited this mutant only for the STD-NMR screening as an artificial dimeric construct that does not exist in nature. We have modified the text to clarify this point, line numbers 532 – 538 (Discussion section).

5. The DSF data need to have their statistical reproducibility stated, for example SEM for the Tm. Plus, significance should be clarified, ie is a 2°C difference significant? Optionally, is this destabilization dose-dependent?

We thank the reviewer for pointing this out. We have revised the data including SEM in Figure 1 (page 13) and Figure S3 (page. S9). Moreover, the statistical analysis can be now seen in Table S8 (page S12, Supporting information).

6. Finally, the start of screening strategy rely on the V190W mutant that is dimeric and you are expecting to identify fragments that bind the very same interface that was mutated. Would not the mutation by a Trp be problematic, ie induce false binding to aromatic compounds? A double control, perhaps with another dimeric mutant on another position would clarify this point and strengthen the NMR screening.

We agree with the reviewer that further elaborating on this point would be helpful. Indeed, the reviewer's suggested mechanism may be the reason behind the identification of 30 fragments that show differential binding between V190W PfMDH and the wild type, but do not show evidence of deep binding. The text has been modified to clarify this point, line numbers 534 – 538 (Discussion section). Further, we have specified what our interpretation of the STD-NMR data can tells us in line numbers 316-317, section 4.1.

7. Section 6.6. Does some of the compounds induce no effect on the thermal stability? Because it would be strange for even compounds like 2b or 6b that are negative for enzymatic activity to display thermal destabilization.

Due to the small size of the fragments we report we cannot exclude the possibility for non-specific binding at other locations of PfMDH. As these fragments are highly hydrophobic in nature it is not unreasonable to suggest that they may negatively impact the protein fold, without necessarily impacting the dimer interface that we have shown is essential for supporting catalytic activity (Lunev et al 2018). We have modified the text to reflect this (line numbers 566 - 570).

8. Section 6.7, line 471-474. The statement of the inhibitory nature of 4PA would need to show the dose-dependency on the enzymatic activity. If higher doses are not possible, lower doses would at least show the disappearance of the effect.

We thank the reviewer for pointing this out. We have revised Figure 5a at page 22 showing now the dose/dependency effect of 4PA at lower concentration of the compound. The corresponding result description can be found in line numbers 792-796, paragraph 4.7. The related data are available in the excel file "4PA inhibition proof.xlsx". Furthermore, Table S6 has been reformatted to address the reviewer comment at page S9, Supplementary information.

9. Section 6.8. I could not understand the statement of this section, what did it bring to the overall study? Plus, compound 2a and 6a are not validated as good analogs of 4DT or 4PA, at least in the data presently displayed.

This observation is correct. We have modified the text to reflect this in line numbers 617 – 631, Discussion section.

Briefly, the 4DT derivatives library was tested with SAXS regardless of activity profile as shown in Figure 4: this would help future compound selection process. As a result, compound **6a** which initially indeed showed low activity profile, opened up the structure of PfMDH as reflected by higher value for radius of gyration 36.4 Å compared to 33.7 Å for apo PfMDH. This means that it did not dissociate the complex of 4 PfMDH subunits, but located itself within the tetrameric assembly, causing them to distance themselves from each other, which is fascinating given the small size of the compound. Furthermore, it showed that SAXS was able to detect compounds otherwise discarded on the sole basis of their biological activity.

#4 Style and formatting

1. “Abbreviations” and “keywords” have been moved between “references” and “conflict of interests”.
2. Word count of the abstract is now 150 words.
3. Bar plot of Figure 4 at page 20 has been converted to dot-plot format in accordance with Formatting guidelines & Policy.
4. The Discussion & Results have been rephrased or rewritten where necessary in order to comply with the 5000 words count guideline.
5. Table S3 has been formatted according to the journal scheme at page S5, Supporting information.
6. Supporting information -> Supplementary information
7. The main text contained an error in the Table count that began with 2 instead of 1 at page 18. The correction can be seen in line numbers #401 and #411 of the revised manuscript.

#5 Other

- Acknowledgment of Dr. Arie Geerlof.
- The data availability statement can be found in paragraph "14, page 32. Furthermore, it specifies that the source of the SAXS data are currently on hold and will only be available upon accepted publication.
- The mention of Table S4 was moved from the "Statistics and Reproducibility" paragraph to "4DT derivative screening"
- Protein and ligand concentration added at line numbers #165 – 166
- The β was missing in ln 141
- μM added in ln 193
- line number 243 -> “compound”
- Line number 452 was removed as the same concept was repeated in the beginning of the section

REVIEWERS' COMMENTS:

Reviewer #3 (Remarks to the Author):

I have carefully looked at the revised manuscript and all my points of concern, as well as those of the other reviewers, have been amended perfectly. It added a lot of clarity to the study and allowed me to have a better understanding of some points, such as the usefulness of SAXS. I believe that it is now more accessible to non-specialist readers.